# Antibiotic Use in Pregnancy: A Global Survey on Antibiotic Prescription Practices in Antenatal Care

**DOI:** 10.3390/antibiotics12050831

**Published:** 2023-04-29

**Authors:** Carlotta Gamberini, Sabine Donders, Salwan Al-Nasiry, Alena Kamenshchikova, Elena Ambrosino

**Affiliations:** 1Institute for Public Health Genomics (IPHG), Department of Genetics and Cell Biology, Research School GROW for Oncology and Reproduction, Faculty of Health, Medicine & Life Sciences, Maastricht University, 6229 ER Maastricht, The Netherlands; c.gamberini@maastrichtuniversity.nl (C.G.); spl.donders@alumni.maastrichtuniversity.nl (S.D.); 2Department of Obstetrics and Gynecology, Research School GROW for Oncology and Reproduction, Maastricht University Medical Centre+, 6229 HX Maastricht, The Netherlands; salwan.alnasiry@mumc.nl; 3Department of Health, Ethics and Society, School of Public Health and Primary Care, Maastricht University, 6229 ER Maastricht, The Netherlands; a.kamenshchikova@maastrichtuniversity.nl

**Keywords:** antibiotic resistance, antenatal care, antibiotic prescription practices

## Abstract

Antibiotic prescription and use practices in the antenatal care setting varies across countries and populations and has the potential to significantly contribute to the global spread of antibiotic resistance. This study aims to explore how healthcare practitioners make decisions about antibiotic prescriptions for pregnant women and what factors play a role in this process. A cross-sectional exploratory survey consisting of 23 questions, including 4 free-text and 19 multiple-choice questions, was distributed online. Quantitative data were collected through multiple-choice questions and was used to identify the most common infections diagnosed and the type of antibiotics prescribed. Qualitative data were gathered through free-text answers to identify gaps, challenges, and suggestions, and the data were analyzed using thematic analysis. A total of 137 complete surveys mostly from gynecologists/obstetricians from 22 different countries were included in the analysis. Overall, national and international clinical guidelines and hospital guidelines/protocols were the most frequently used sources of information. This study highlights the crucial role of laboratory results and guidelines at different levels and emphasizes region-specific challenges and recommendations. These findings underscore the pressing need for tailored interventions to support antibiotic prescribers in their decision-making practice and to address emerging resistance.

## 1. Introduction

Antibiotic use in pregnancy has been globally on the rise in the last decade [1], including during pregnancy [2]. According to previous studies, antibiotics account for approximately 80% of drug prescriptions during pregnancy, with an estimated 20–40% of expectant mothers receiving them across different countries in recent years [3]. Antibiotics are powerful drugs used to treat infections; as such, they have contributed to the saving of countless of lives, including those of pregnant women. Indeed, the unique immunologic and physiologic characteristics of pregnancy are associated with high rates of serious and sometimes fatal outcomes from a variety of infectious diseases [3,4,5]. Untreated infections during pregnancy and delivery are key contributors to maternal and neonatal morbidity and mortality and can largely be prevented by improving the quality of antenatal care (ANC), which aims to be effective, safe, and efficient through judicious use of medications [6,7]. Some of the most common maternal infections are urinary tract infections (UTIs), respiratory tract infections (RTIs), sexually transmitted infections (STIs), bacterial vaginosis (BV), and group B streptococcus infections (GBS) [8,9].

Maternal mortality rates remain high worldwide, with around 300,000 women dying in 2017 according to the World Health Organization (WHO) [10], and global estimates suggest that in 2014, infections were the third leading cause of maternal mortality, or 10.7% of all maternal deaths [11]. In 2019, the WHO reported that around 70 pregnant women per 1000 live births had an infection that required hospitalization. The same report identified that in high-income countries, 11 women per 1000 live births with an infection had adverse maternal outcomes; meanwhile, in low- and middle-income countries, up to 15 women per 1000 births were impacted [10].

Antibiotic treatment is widely accepted as the best way to treat the majority of bacterial infections. Because many antibiotic regimens have similar efficacy, the choice for the type of antibiotic is based on pharmacokinetics, safety, and cost [12]. When considering antibiotic use during pregnancy, antibiotic safety for the mother and fetus is critical because some drugs can be teratogenic or harmful to the developing fetus [13]. Several factors can influence the risk of teratogenicity, including gestational period, dose and duration of therapy, genetic predisposition, environmental factors, and the degree of drug transfer across the placenta [14,15]. Antibiotic prescriptions should therefore be carefully assessed on an individual basis, comparing the advantages and the risks to both the fetus and mother. In certain instances, antibiotic use has been linked to an increased prevalence of neonatal necrotizing enterocolitis, while in others it has been linked to a lower rate of lung problems and serious cerebral abnormalities when compared to non-antibiotic-treated mothers [16].

While antibiotics are essential drugs for the treatment of infectious diseases, data on antibiotic efficacy and safety during pregnancy are extremely limited, including the gestational age they are consumed. In part, this is due to legal and ethical concerns restricting research on pregnant women [17,18]. Typically, clinical and epidemiological studies on drug safety are conducted on non-pregnant women, with results extrapolated to pregnant women [19]. Only 10% of pharmaceuticals sold since 1980 are thought to have collected appropriate data on risks when used in pregnancy, while over 98% of medications lack adequate pharmacokinetic or safety data on dosing in pregnant women [20].

In the context of global maternal and childcare, the selection of optimal antibiotics, their dosages, the duration of therapy, and the balance between costs and benefits are crucial factors to acknowledge. Guidelines play an important role in ensuring that antibiotics are administered correctly. Schuts et al. [21] described the relationship between prescribing antibiotics according to guidelines and a significantly lower risk of mortality in hospitals overall. In addition, compliance with clinical practice and guidelines is an indicator of high-quality treatment in hospitalized patients [22,23]. Although global attention to ANC therapeutic needs is gradually increasing, there are still gaps, primarily in knowledge, that undermine the development or attainment of guidelines to direct physicians’ practices.

Adequate antibiotic prescription is crucial to ensure that the right patient receives the right antibiotics at the right time in order to optimize clinical outcomes, while also helping to limit further increases in antibiotic resistance. In fact, as antibiotics reduce the risk of serious morbidity and mortality in a population, they also facilitate the evolution and spread of antibiotic-resistant bacteria, and antibiotic over-prescription hastens this process. Pregnant women and newborns require special consideration. As such, understanding how these issues overlap is urgent, and investigating the role that ANC providers play is empirical. How and when providers prescribe drugs, including antibiotics, is a multifactorial and complex process. Previous research has identified a variety of factors influencing antibiotic prescriptions in hospitals and primary care, including physician-specific and patient-related factors, including the availability of diagnostic tools, local antibiotic resistance data, patient satisfaction, and cultural and organizational factors.

This study investigated how ANC providers in various countries make decisions about antibiotic prescriptions for pregnant women. Improved understandings of antibiotic prescription practices can help healthcare providers better tailor their practices to the specific needs of patients, resulting in improved maternal and neonatal health outcomes. By suggesting evidence-based recommendations for ANC providers, this study can help explore current practices and needs for improvements at the policy and institutional levels to support antibiotic prescribers in their decision-making processes and improve the consistency of care. This, in turn, can lead to more reliable outcomes and a more evidence-based approach to healthcare management and policy development. By having a better understanding of the factors that influence antibiotic prescription decisions, healthcare managers can develop targeted interventions that improve the quality of care and patient outcomes.

## 2. Results

### 2.1. Respondents’ Characteristics

In total, 161 survey responses were received. However, 24 survey responses were excluded from further analysis as they had missing answers on more than 90 per cent of the questions. In total, 137 healthcare professionals participated in this study, representing 22 different countries from four WHO regions: the European Region (8); the Eastern Mediterranean Region (2); the African Region (7); and the Region of the Americas (5), of which there were several from Central and South America and one respondent from Canada. See Appendix C for a map of the geographical distribution of the survey. Most of the respondents were gynecologists/obstetricians (84.7%). Other professions included gynecologists in training (5.1%), general practitioners (3.6%), public health professionals, and one antimicrobial resistance (AMR) fellow (1.5%) (Table 1).

### 2.2. Practices of Antibiotic Prescription

The main themes and factors that influenced antibiotic prescribing practice in the study population are summarized in Figure 1.

#### 2.2.1. Diagnostics

Bacterial infections were the most diagnosed infections (81%) across all WHO regions. Fungal infections were only reported (10.9%) in the European and Eastern Mediterranean regions.

Participants from the European and Eastern Mediterranean Regions reported diagnosing infections more than once per month, those from the African Region at least once a week, and from the Americas both routinely and regularly. Most of the respondents (60.6%) diagnosed infections in pregnant women more than once a month. A little over a third of all respondents (35.8%) diagnosed infections more than once a week. In the African Region, half of the respondents indicated that they diagnosed an infection more than once a week.

In this study, laboratory diagnostics and clinical presentation were the most frequently considered factors for diagnosing suspected infections in pregnant women across all four WHO regions. Regarding laboratory access, the majority of respondents (71.5%) reported having easy and prompt access to laboratory diagnostics. However, regional discrepancies were observed, with 50% (4 out of 8) of the respondents from the African Region indicating partial access and 37.5% (3 out of 8) reporting easy access.

Within the African Region, specific challenges were identified in Zimbabwe, where adequate laboratory support was lacking, as well as in Nigeria, where delays in receiving laboratory investigation results were reported. In Uganda, respondents reported non-adherence to ministry guidelines on antibiotic use for pregnant mothers, lengthy laboratory investigation times, and a reliance on clinical presentations to make diagnostic judgments. It is unclear whether similar challenges exist in other countries, and further investigation is warranted to contextualize these findings.

In the Americas, 56.3% (9 out of 16) of respondents reported easy access to laboratory diagnostics, while the remaining 6 respondents reported partial access. There was no discernible difference between North and South America in this regard. On the other hand, all four respondents who reported difficult access to laboratory diagnostics originated from the European Region, and all were gynecologists/obstetricians from Italy. These findings may have important implications for the development of region-specific interventions to improve laboratory access and facilitate appropriate antibiotic prescribing practices. National clinical guidelines were also frequently taken into consideration (52.6%), as well as previous medical history (38.0%). One respondent from the American Region mentioned lifestyle and diet as additional elements taken into consideration during diagnosis.

#### 2.2.2. Treatment

Across the four WHO regions, the most common infections that required antibiotic treatment were UTIs (67.9%). Other infections requiring antibiotic treatments included BV (5.8%), candidiasis (4.4%), asymptomatic bacteriuria (2.2%), and vaginitis (2.2%). In another 5.8% of cases, the free-text response of *Escherichia coli* infections was provided. It is important to note is that *Escherichia coli* infections are often associated with UTIs and therefore likely contribute to the response chosen most often.

The majority of respondents across all four WHO regions indicated prescribing broad-spectrum antibiotics more often (56.9%) than narrow spectrum (31.4%). Data showed that out of the 34 respondents who indicated partial or difficult access to laboratory diagnostics, 24 (70.6%) indicated prescribing broad-spectrum antibiotics most often, versus 6 respondents who indicated prescribing narrow-spectrum antibiotics more often.

Just over a quarter of the total respondents (28.5%) never referred pregnant women with a suspected infection to a colleague, while over half rarely did. Out of 14 respondents who regularly/routinely referred patients to another specialist, 9 (64.3%) were gynecologists/obstetricians, but only 5 out of 21 (23.8%) of respondents with a different occupation referred patients regularly/routinely. These 5 included a nurse (1), lab technician/AMR fellow (1), midwife (1), gynecologist in training (1), and general practitioner (1)

In case of referral, this was most often carried out by an infectious disease specialist (46%). Other patients were referred to gynecologists (12.4%), female urologists (5.1%), and pathologists (0.7%).

#### 2.2.3. Indicators Considered when Prescribing Antibiotics

Different factors were considered by practitioners when prescribing antibiotics to pregnant women, including laboratory results (73.7%), overall clinical picture (46.7%), trimester of pregnancy (43.1%), national clinical guidelines (40.1%), and international clinical recommendations (35.8%). There were, however, some differences across the four WHO regions. National clinical guidelines were indicated as more important by the European Region (44.6%) and the Americas (43.8%) compared to the African (25.0%) and the Eastern Mediterranean (8.3%) Regions. Additionally, international clinical guidelines were not mentioned by respondents from the African Region.

#### 2.2.4. Consultations

Respondents were asked more specifically about the frequency of consultations with colleagues. About two thirds of all respondents (61.3%) consulted with colleagues before prescribing antibiotics to a pregnant woman, whereas about one quarter (29.9%) indicated that they never did. This trend was visible across almost all age groups, 20–29 years of age (66.7% vs. 8.3%), 30–39 years of age (76.9% vs. 15.4%), and 40–49 years of age (59.1% vs. 27.3%). Only in the age group of 50+ was the proportion of respondents consulting with colleagues similar to those who did not (48.3% vs. 45.0%).

When consulting with colleagues, respondents mentioned doing so most often with a gynecologist (33.6%) or a microbiologist (32.1%). Other colleagues mentioned were infectious disease specialists and midwives. Most of the respondents who consulted with colleagues did so rarely (38.7%). In the American, European, and African Regions, around twenty per cent of the respondents consulted with colleagues more than once a month, whereas only one of the respondents from the Eastern Mediterranean Region consulted more than two to three times a year with colleagues. Overall, only a small number of respondents (4.4%) consulted weekly with colleagues.

#### 2.2.5. Reasons for Switching between Hospital and National Guidelines

Among the respondents, 22 reported instances of non-adherence to hospital and national guidelines. Reasons for not following guidelines varied across regions. In Europe, reasons included concerns about allergies, antibiotic resistance, and unclear policies. In Switzerland, antibiotics were used cautiously, and alternative strategies were discussed with patients whenever possible. In Italy, antibiotic resistance was a primary concern, while allergies were cited as a reason for non-adherence in Belgium and Lithuania. In the Netherlands, second or third choice antibiotics were used in cases of allergy or resistance based on guidelines or consultation with a microbiologist. However, the guidelines themselves were sometimes unclear or subject to discussion, particularly in cases of preventive antibiotics, such as with preterm rupture of membranes. For instance, a respondent from the Netherlands explained the challenges related to guidelines use in pregnancy in an open question:

“Particularly with preventive antibiotics, there is sometimes discussion in guidelines. For example, in case of preterm ruptured membranes. Previously, the advice was to treat, then not for a while, and now there is a review in which the option is considered again. This is based on the same trials. So, it is not always clear what the best policy is.”(The Netherlands; translated from Dutch.)

Respondents from the Americas mentioned natural remedies and non-response to antibiotics from guidelines as reasons for deviation. Based on the experience from the respondent from Canada, clients were supported to make informed choices and if they chose to try natural remedies or aromatherapy first, antibiotics were used only if the first treatment was not successful. In Uruguay, deviation from guidelines occurred sometimes due to non-response to the proposed treatment. In the Eastern Mediterranean Region, some respondents reported deviation due to personal experiences and lack of response to antibiotics. In Iraq and Lebanon, treatment decisions were made based on culture results. In the African Region, respondents mentioned resistance and drug availability as reasons for deviation. In Zimbabwe, overuse of Ceftriaxone led to resistance. An example of a respondent from Nigeria has provided insight into the difficulties surrounding medication and drug availability by responding to an open-ended question.

“Working in rural areas within LMICs (low and middle-income countries) often requires prescriptions based on available medications amongst other considerations.”(Nigeria.)

It is important to note that the reasons for deviation may differ based on the healthcare setting and not just because of overuse or misuse of antibiotics. Therefore, guidelines should be contextualized according to the specific setting to ensure appropriate and effective use of antibiotics.

### 2.3. Challenges of Diagnostics/Antibiotic Treatment

Twenty-eight respondents (22.4%) reported encountering challenges related to the diagnosis and/or prescription of antibiotics for pregnant women in their daily practice, with variations across different WHO regions. Many respondents from the African Region (71.4%) reported facing such challenges, whereas in the European Region and Eastern Mediterranean Region, a relatively smaller proportion of respondents (11.5% and 20.0%, respectively) reported facing similar challenges. In the Region of the Americas, nearly half of the respondents (42.9%) reported encountering such challenges. In Europe, challenges related to the diagnosis of infections and/or antibiotic prescription for pregnant women included inadequacy of antibiograms, allergies, and resistance. Specifically, respondents from Switzerland identified challenges related to atypical pneumonia, preterm labor, preterm rupture of membranes, appendicitis, and asymptomatic UTIs. In Italy, inadequate use antibiograms were reported.

“Antibiograms often investigate antibiotics that cannot be prescribed during pregnancy”(Italy; translated from Italian.)

In the Netherlands, challenges included unknown allergies to antibiotics and the need to switch medications due to lack of clinical improvement, as well as a lack of studies on proper dosage for pregnant women and the degree of transmission to the child.

“- Not unequivocal best policy—There is a lack of studies on proper dosing for pregnant women and on the extent of transmission to the child. During delivery, leucocytes and CRP are routinely elevated, which makes differentiation between infection and inflammation due to delivery not always possible.”(The Netherlands; translated from Dutch)

In the Eastern Mediterranean Region, poor compliance and disagreement pose challenges. In Iraq, poor compliance and limited investigations available, or the unavailability of certain medications, were identified as issues. In Lebanon, resistance patterns present in patients and conflicting laboratory test results that do not fit with symptoms pose challenges that require collaboration and resolution.

In the African Region, challenges included lack of lab support, delays in lab results, resistance, and recurrent UTI. Respondents from Uganda reported not adhering to ministry guidelines on antibiotic use for pregnant women due to long delays in lab investigations. Zimbabwe faces a lack of adequate laboratory support, while respondents from Nigeria experience delays in laboratory investigations. In Ethiopia, selecting antibiotics that are not associated with congenital malformations in case of resistance development poses a challenge in the first trimester of pregnancy. In Liberia, pregnant women presenting with recurrent UTI were identified as a challenge that requires urine culture and sensitivity tests. It is important to note that these findings are based on the experiences reported by our respondents in Ethiopia and Liberia and may not be representative of the entire countries.

In the Americas, challenges included resistance, sensitivity, lack of consideration for diet, and lifestyle factors. Specifically, respondents from the Dominican Republic reported challenges when faced with sensitivity to antibiotics not recommended for the appropriate trimester of pregnancy in urine or vaginal culture reports. In Uruguay, challenges include uncertainty around whether patients are immunized or experiencing toxoplasmosis reinfection, as well as multi-resistant germs that require treatment with antibiotics not ideal in pregnancy.

“When I have a report of a urine or vaginal culture that gives me sensitivity to antibiotics not recommended for the corresponding trimester of pregnancy.”(Dominican Republic; translated from Spanish.)

“There are multi-resistant germs that make it necessary to treat with antibiotics that are not ideal in pregnancy.”(Uruguay; translated from Spanish.)

### 2.4. Sources of Information

The results indicate that national clinical guidelines (67.1%), international clinical guidelines (63.5%), and hospital guidelines/protocols (56.2%) were the most frequently used sources of information among healthcare professionals. Professional education courses were also identified as a commonly used source of information (42.3%). However, there were differences in the sources of information used between the four WHO regions. In Europe, national clinical guidelines (71.3%), international clinical recommendations (61.4%), and hospital guidelines (60.4%) were the top three sources of information. In the Eastern Mediterranean Region, international clinical recommendations (58.3%), professional conferences (41.7%), and hospital guidelines (41.7%) were the most used sources. In Africa, where there is limited resource availability, healthcare professionals heavily rely on professional education courses (87.5%), followed by international clinical recommendations (75.0%) and national clinical guidelines (62.5%). In the Americas, both national clinical guidelines and international clinical recommendations were equally used (75.0%), while hospital guidelines were used by 50.0% of healthcare professionals.

### 2.5. Recommendations

The respondents’ recommendations indicate an urgent need for support in the form of guidelines, access to resources, and training manuals to aid in decision making regarding the diagnosis and treatment of infections in pregnant women. The reduction in antibiotic use was suggested in Canada and Belgium, while considerations for resistance when prescribing were recommended in Uruguay and Italy. Italy also called for more specific guidelines for resistant bacteria. More uniform and better guidelines for prescription during pregnancy, increased education for healthcare professionals and patients, and improved lab accessibility were also recommended.

“Taking into account antibiotic resistance. Taking into account the unavailability in Italy of amoxicillin for prevention of Streptococcus agalactia infection in labor.”(Italy; translated from Italian.)

“Publish updated national/regional pocket guidelines based on local epidemiology. Establish a national registry on (severe) pregnancy infections treated in inpatient settings.” (Italy; translated from Italian.)

Several specific recommendations were provided by different countries, such as non-antibiotic therapy in pregnant women in Switzerland, utilizing culture and sensitivity testing by clinicians in Uganda and Belgium, limiting the use of antibiotics and prioritizing the results of the antibiogram in the Netherlands, and refraining from prescribing first-line antibiotics, which are the standard antibiotics that are typically prescribed as a first course of treatment for a particular infection, without a proper diagnosis, in Mozambique.

“Hospital develop antibiograms for mothers and should be adhered to. Culture and sensitivity testing should be fully utilized by the clinicians.”(Uganda.)

Respondents from Uruguay suggested better treatment of UTIs during pregnancy, along with following national and international clinical guidelines and avoiding prophylactic antibiotic use.

Respondents from Italy recommended teratology courses, establishing a national registry of hospitalized infections in pregnancy. Considering the risk of AMR and the limited availability of antibiotics, it may not be advisable to rely solely on antibiotic treatment to prevent GBS infection. More specificity of therapy and options in case of resistance, as well as easily accessible guidelines, were also recommended in Italy. In Lebanon, teaching patients and teamwork were suggested, while upgrading laboratory services, especially culture and sensitivity, was recommended in Liberia.

Respondents from the Dominican Republic emphasized the importance of following guidelines, using logical judgment, and continually updating medical practices. Finally, respondents from Canada recommended considering treatment with natural remedies before antibiotics, if appropriate and suitable for the patient.

## 3. Discussion

The primary objective of this study was to present a comprehensive overview of diverse antibiotic prescription practices for pregnant women among healthcare professionals from different regions of the globe.

In the context of diagnosing and treating infections in pregnant women, most healthcare professionals reported engaging in such practices as part of their routine clinical activities. Despite considerable similarity in responses across respondents from various WHO regions, there were notable differences observed, specifically with respect to the accessibility of laboratory diagnostic testing. Respondents from the Regions of the Americas, Europe, and Eastern Mediterranean commonly reported easy and rapid access to laboratory diagnostic testing, whereas others reported limited access. However, caution is needed when generalizing the findings, as the percentage of those reporting limited access was low in all regions. The challenges faced by respondents from the African Region in accessing laboratory diagnostic testing were also reflected in issues related to prescribing antibiotics. These disparities in laboratory diagnostic accessibility align with the concept of the global diagnostic gap, which describes the unequal distribution of diagnostic resources worldwide [24]. Individuals residing in larger urban areas or with higher socioeconomic status generally enjoy better access to laboratory diagnostic testing, whereas those from rural areas or with lower socioeconomic status are disproportionately affected [25]. Insufficient access to high-quality laboratory diagnostic testing can have significant implications for pregnant women’s care, potentially leading to inaccurate diagnoses and inappropriate treatment regimens [26]. Such circumstances may also contribute to the emergence of antimicrobial resistance.

Notably, UTIs were the most treated infections across all four WHO regions, which is consistent with previous research that identifies UTIs as the most prevalent infections among pregnant women [27]. The majority of respondents reported prescribing broad-spectrum antibiotics most frequently, which aligns with empirical treatment practices observed in the general population. However, research has also shown that once laboratory diagnostic results are obtained, clinicians often shift to prescribing narrow-spectrum antibiotics [28]. This pattern was not reported in the present study. Additionally, some respondents discussed the applicability of guidelines and the potential adverse clinical outcomes of prescribing certain antibiotics during pregnancy, particularly in specific trimesters.

A few respondents In our study also mentioned the inadequacy of antibiograms for infections during pregnancy. According to them, antibiograms often fail to evaluate antibiotics that are safe to use during pregnancy, while some of the evaluated antibiotics are contraindicated during pregnancy due to their teratogenic effects [29]. Moreover, resistance to commonly used antibiotics among prevalent bacterial strains is widespread, limiting the choice of antibiotics during pregnancy. Although β-lactam antibiotics are generally considered safe during pregnancy, resistance rates to these antibiotics are rapidly increasing [30]. Furthermore, little is known about the effects of many antibiotics on maternal and fetal health. A study reported that the vast majority of antibiotics approved by the Food and Drug Administration (FDA) had an ‘undetermined’ potential to cause fetal abnormalities [31]. This lack of evidence-based information on the safety and efficacy of most drugs during pregnancy has resulted in insufficient support for decision-making processes concerning antibiotic prescriptions for pregnant women, as described by other studies [32,33]

In terms of sources of information, variations were observed among the four WHO regions in our study. National clinical guidelines, international clinical guidelines, and hospital guidelines/protocols were the most frequently consulted sources of information overall. However, in the African Region, professional education courses were the most used source, while professional conferences and education courses were among the top three sources in the Eastern Mediterranean Region. The lower utilization of national and institutional clinical guidelines in the African Region may be attributed to the limited availability of standardized treatment guidelines in many African Union member states [34]. Similarly, the lack of clear guidelines regarding dosage and duration of antibiotic treatment in Lebanon has been identified as a potential reason for the low reliance on national and institutional guidelines in the Eastern Mediterranean Region. These findings suggest that healthcare professionals follow established guidelines and training procedures that prioritize diagnostic accuracy in identifying infections [35]. However, it is important to note that the specific guidelines or training protocols followed by healthcare professionals may vary depending on their location, setting, or specialty. Therefore, it may be beneficial to further explore the underlying factors that contribute to these diagnostic practices and to consider potential improvements or modifications to existing guidelines or training programs to ensure optimal diagnosis and treatment of infections in pregnant women.

The survey also revealed that approximately one third of the respondents consulted colleagues before prescribing antibiotics to pregnant women, with gynecologists and microbiologists being the most frequently consulted. Previous research identified four key characteristics of consultation with colleagues, including the influence of colleagues, social team dynamics, hierarchy, and reputational risk [36]. However, our study did not explore the specific motivations for consulting with colleagues among the respondents. The process of referral and consultation can vary significantly across different countries and healthcare systems. Referral and consultation mechanisms may differ based on factors such as the organization of healthcare systems, the availability of specialists, and the criteria for referral [37]. For instance, in some countries, patients may require a referral from a primary care physician before seeing a specialist, while in others, patients may be able to directly access specialists without a referral. Additionally, the process of consultation may involve various methods of communication, such as telephone, electronic messaging, or face-to-face meetings. These differences in referral and consultation processes may have implications for the quality and timeliness of care that patients receive, and healthcare professionals may need to adapt their practices based on the specific systems in which they work. One possible explanation for the trend in higher proportions of respondents consulting with colleagues among younger age groups compared to older age groups could be the difference in experience and confidence levels in their respective fields. Younger individuals who are newer to their professions may be more likely to seek guidance from colleagues to gain knowledge and build their confidence, while older individuals with more experience may have already developed a more independent decision-making process. However, further research would be needed to confirm this hypothesis.

The prescription practices of antibiotics can vary greatly between countries due to differences in regulatory frameworks, healthcare systems, and cultural norms. For instance, some countries have more stringent regulations regarding antibiotic use, which may result in lower rates of prescription and a greater emphasis on non-antibiotic treatments. Several countries have implemented strategies to promote appropriate antibiotic use, resulting in lower rates of antibiotic use and a greater emphasis on non-antibiotic treatments. For example, in the Netherlands, there is a strong emphasis on responsible antibiotic use, which has led to lower rates of antibiotic use in the general population [38,39]. In Canada, initiatives focused on education and awareness have been implemented to promote appropriate antibiotic use, resulting in more conservative prescribing practices compared to the United States [40]. Contrary to the conventional assumption that behavioral factors, such as education, are the main drivers of antibiotic practices, recent research suggests that these practices should be examined from the perspective of infrastructural constraints. In other words, systemic factors, including the availability of healthcare facilities, drug supply chains, and laboratory diagnostic capacity, are important determinants of antibiotic use, in addition to individual-level factors [41,42]. While some countries may have more relaxed regulations that encourage the use of antibiotics, leading to higher rates of prescription, there are also systemic factors that contribute to inappropriate antibiotic use. For instance, in Nigeria, there may be infrastructural challenges, such as limited access to healthcare facilities and qualified medical personnel, which further exacerbates the overuse and misuse of antibiotics [43]. Similarly, in Uganda, infrastructure challenges, such as inadequate drug supply chains, limited laboratory diagnostic capacity, and insufficient staffing also contribute to inappropriate antibiotic use [44]. In Italy, systemic factors such as inadequate prescribing guidelines and a lack of antimicrobial stewardship programs may also play a significant role in driving inappropriate antibiotic use [45,46].

The differences in prescription practices may have significant implications for the emergence and spread of antibiotic resistance, as well as for patient outcomes. Currently available guidelines and support were evaluated by the majority of the respondents in our study, who appeared either completely or somewhat satisfied. However, some respondents expressed some dissatisfaction, and provided recommendations for improving the diagnostics and treatment of pregnant women with bacterial infections. The findings are consistent with previous research, which has shown an improvement in AMR awareness among healthcare professionals over time [36,47]. Despite this, our respondents highlighted a need for better guidance and support for the treatment of resistant bacteria in pregnant women. To address this need, it is recommended that local guidelines are developed specifically for the most frequently occurring (resistant) bacterial strains.

Moreover, the respondents in this study exhibited an awareness of the importance of careful use of antibiotics, particularly in cases of prophylaxis, as a means of curbing the emergence of antibiotic resistance. Research conducted previously has indicated that antibiotics are often administered routinely to women in labor in low- and middle-income countries, regardless of the complexity of delivery [48]. This practice of prophylactic antibiotic use could have a detrimental effect on the emergence of antibiotic resistance. Therefore, raising awareness about safe and effective antibiotic use could help prevent antibiotic resistance. Additionally, a few respondents identified the need for high-quality and easily accessible laboratory tests to ensure appropriate antibiotic prescribing. As Kollef [28] suggested, initial broad-spectrum antibiotic treatment should be narrowed or adjusted based on the results of culture and sensitivity tests to minimize the risk of death due to ineffective initial treatment, as well as the risk of antibiotic resistance resulting from prolonged use of broad-spectrum antibiotics.

An online survey was utilized to collect data for this study due to its global scope. However, this method may have introduced sampling bias. The decision to participate in the survey could have been affected by access to the internet and technological proficiency, potentially dissuading certain medical professionals from contributing to the study, particularly those in areas with limited internet connectivity. This could have led to distorted survey results, particularly in relation to information sources used for diagnostics and antibiotic treatment. One potential limitation is the relatively low number of responses received despite our international recruitment efforts. We recognize that a larger sample size would have provided a more representative and diverse range of perspectives on the determinants of antimicrobial use and resistance; the limited number of responses should be considered when interpreting the findings of our study. Another limitation of our study is that we did not collect specific information on the names or families of the antimicrobials used in our sample. As a result, we were unable to classify the antimicrobials according to the WHO AWaRe classification (WHO access, watch, reserve, classification of antibiotics for evaluation and monitoring of use), which could provide valuable information on their appropriateness and potential impact on antimicrobial resistance.

Furthermore, this study is limited by significant variations in the number of respondents across the different WHO regions. Many of the respondents were from the European Region, with many of those respondents originating from Italy. Some of the researchers’ Italian background may have resulted in more responses from Italy to the post they shared on social media, and snowball sampling may have compounded this effect. As a result, the study’s findings may not be representative of the global population. Finally, while acknowledging the diversity of gender identities among patients, the present study employs the term ‘pregnant women’ to refer to this population. This choice is justified on several grounds. The term ‘pregnant women’ is widely used in ANC facilities worldwide. Therefore, it was deemed appropriate to adopt this terminology consistently in the survey and subsequent paper. Notably, this terminology does not preclude individuals identifying as a different gender to be included.

## 4. Materials and Methods

A cross-sectional online survey was carried out among healthcare providers working in ANC from different countries around the globe between 1 June 2022 and 1 January 2023.

### 4.1. Online Survey

The online survey consisted of 23 questions, including 4 free-text answers and 19 multiple-choice questions. It focused on three aspects: practices of antibiotic prescription, sources of information, and recommendations for future practices. The survey was first developed in English, after which it was shared with a multidisciplinary group of healthcare professionals for feedback and to ensure the quality of the questions. The finalized survey was translated with the support of native speakers into six more languages: Dutch, German, French, Spanish, Portuguese, and Italian. A further quality check was performed to ensure the content was consistent across translations and to ensure the absence of grammatical or spelling errors. The survey was generated online using Qualtrics XM (Qualtrics, Provo, UT, USA, 2022). See Appendix A for the survey.

### 4.2. Study Population

The study population consisted of healthcare practitioners working in ANC with no geographical limitation. To reach a broad number of participants, a combination of purposive sampling and snowball sampling was used. Participants were recruited on a voluntary basis and were invited to participate via email, by contacting professional organizations/networks, or by sharing the information on social media platforms (LinkedIn, Twitter, and Facebook). Email addresses of practitioners or networks were gathered from the main researchers’ personal networks, as well as the International Federation of Gynaecology and Obstetrics (FIGO) website, which was utilized to compile an email list of member countries. The Gynécologie Suisse SSGO (Swiss association of gynecology) shared the project information on their monthly newsletter. The email and an information leaflet (in each of the seven languages) included a link to the online survey, a brief explanation of the project, its major goals, and the researchers’ names and contact information. Participants were also invited to share the email and information flyer with their professional networks to recruit more participants.

### 4.3. Data Analysis

Quantitative and qualitative data on prescription practices and decision-making tools were obtained using closed and open-ended questions. Data were utilized to identify the key ideas that influence antibiotic prescribing behavior. Quantitative data were gathered and analyzed using descriptive statistics (frequencies and percentages) in IBM SPSS Statistics 27 (Armonk, NY, USA: IBM Corp). Because of the heterogeneous number of responses across countries, data were reported per WHO region.

To generate common themes from qualitative data, descriptive thematic analysis of free-text responses was used. DeepL Translator (DeepL Translate, n.d.) was used to translate free-text responses to English, and the translations were reviewed by native speakers. Initially, a deductive approach was utilized to develop general themes based on survey question topics. Following that, an inductive approach was used to discover additional (sub)themes based on responses. A coding technique was established after reading the answers several times. An overview of the main themes and factors was developed (Figure 1).

In the context of antibiotic prescribing practices, the term “rarely” was used to indicate a frequency of 2 to 3 diagnoses per year, while “regularly” denoted a frequency of more than 1 diagnosis per month, and “routinely” signified a frequency of more than once a week. These terms were used to describe the frequency with which healthcare providers diagnose infections and subsequently prescribe antibiotics.

### 4.4. Ethical Considerations

The study received ethical approval from Maastricht University Medical Ethics Review Committee (FHML- REC/2022/001). Informed consent was obtained from each respondent prior to participation in the survey (Appendix B). The participants had to agree online before going forward with the survey, and they were informed that they could stop at any time by closing the browser. Participants were also informed about the purpose of the study, that participation was voluntary, and that all data collected would be confidential.

## 5. Conclusions

This study sheds light on the decision-making process underlying antibiotic prescription among healthcare practitioners in ANC globally. Data suggest that consultation of laboratory test results and adherence to guidelines were among the most important factors in this process. The study also identified differences in practices and challenges across four WHO regions, indicating the need for region-specific guidance. Furthermore, this study highlights the need for continued efforts to prevent the emergence of AMR within ANC through the implementation of effective and safe antibiotic use practices. This exploratory work provides important insights that can be used to inform the development of targeted interventions and policies to improve the care of pregnant women worldwide. This study can stimulate further research on this topic, including studies on the effectiveness of current guidelines and the development of better guidelines for antibiotic prescription to pregnant women. Research should also investigate the significance of patient expectations in the decision-making process of healthcare practitioners regarding antibiotic prescriptions as this can have implications for the emergence and spread of AMR. Lastly, this study can be beneficial to researchers working on global health issues as it highlights the importance of identifying challenges and regional differences in the implementation of guidelines for antibiotic prescription in ANC.

## Figures and Tables

**Figure 1 antibiotics-12-00831-f001:**
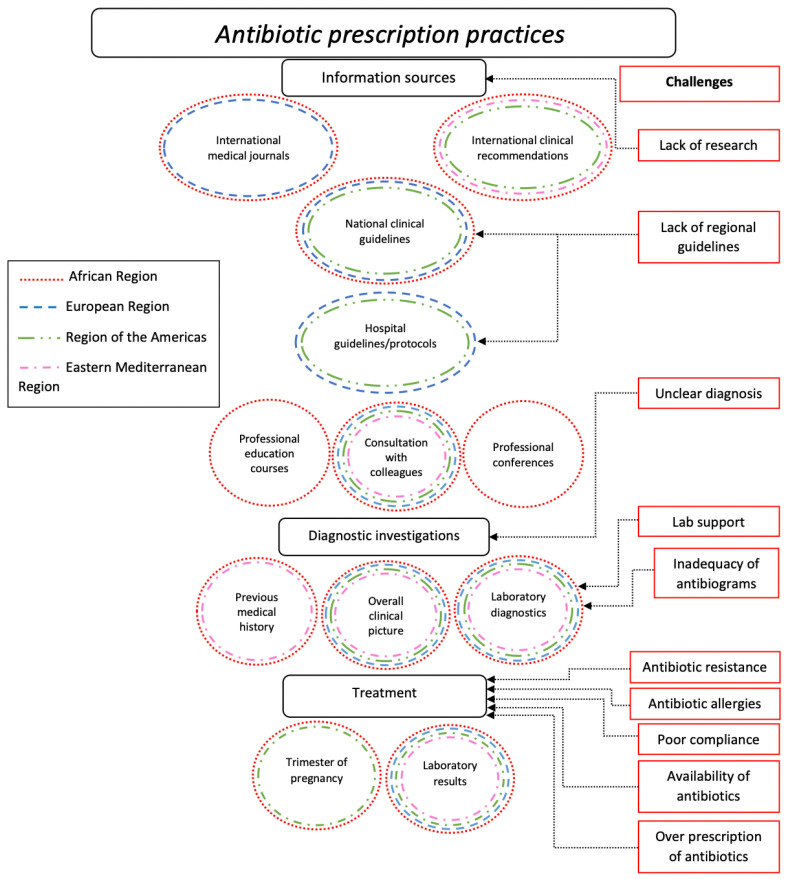
Visual representation of the most common reasons for prescribing antibiotics and the factors that influenced providers’ decision making in each case and the challenges described.

**Table 1 antibiotics-12-00831-t001:** Summary of the main characteristics of healthcare providers who participated in the survey (n = 137).

Characteristic	Total (N = 137), n (%)
Gender	
Male	44 (32.1)
Female	93 (67.9)
Non-binary	0 (0)
Prefer not to say	0 (0)
Age	
20–29	12 (8.8)
30–39	39 (28.5)
40–49	22 (16.1)
50+	60 (43.8)
Other	4 (2.9)
Profession	
Gynecologist/obstetrician	116 (84.7)
General practitioner	5 (3.6)
Midwife	2 (1.5)
Nurse	1 (0.7)
Other	13 (9.5)
Country of practice (WHO region)	
Region of the Americas	16 (11.7)
European Region	101 (73.8)
African Region	8 (5.8)
Eastern Mediterranean Region	12 (8.8)

## Data Availability

Data are contained within the article.

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
