# Peer review of "Antibiotic Use in Pregnancy: A Global Survey on Antibiotic Prescription Practices in Antenatal Care"

_antibiotics, 2023, doi:10.3390/antibiotics12050831_

Round 1

Reviewer 1 Report

This is a critical study addressing antibiotic use in pregnancy. However, the following issues need to be addressed before considering this manuscript for publication:

·       Lines 133 – 139: The authors used two different forms of decimal points: “10.9%” in line 133 and “60,6%” in line 138, for example. They need to use the correct form of the decimal point and be consistent throughout the manuscript.

·        Line 170: The authors mentioned “Escherichia Coli infections” among the “other infections requiring antibiotic treatments.” However, E. coli can cause different types of infections, including UTIs. Therefore, it would be inaccurate to combine them into one category.

·       The authors referred to Figure 1 on page 4, but the figure was included on page 7. Including the figure close to where it was first mentioned is better. Also, the figure's alignment with the text was incorrect since some of the main text was included inside the figure.

Author Response

Comments and Suggestions for Authors (Reviewer 1)

This is a critical study addressing antibiotic use in pregnancy. However, the following issues need to be addressed before considering this manuscript for publication:

A: Thank you for recognizing the relevance of our work. We would like to overall thank the reviewer for taking the time to go over our manuscript and provide useful and constructive feedback.

  • Lines 133 – 139: The authors used two different forms of decimal points: “10.9%” in line 133 and “60,6%” in line 138, for example. They need to use the correct form of the decimal point and be consistent throughout the manuscript.

A: Thank you for pointing this out, this has been correctly formatted in lines 133-139 and throughout the manuscript.

  • Line 170: The authors mentioned “Escherichia Coli infections” among the “other infections requiring antibiotic treatments.” However, E. coli can cause different types of infections, including UTIs. Therefore, it would be inaccurate to combine them into one category.

A: “Escherichia Coli infections” was a free answer that was given from a survey respondent when asked what the most common infection were managed in pregnant women. This has been clarified in the manuscript, lines 208-211.

  • The authors referred to Figure 1 on page 4, but the figure was included on page 7. Including the figure close to where it was first mentioned is better. Also, the figure's alignment with the text was incorrect since some of the main text was included inside the figure.

A: Thank you for pointing this out, Figure 1 has been moved closer to the first mention in page 4. Text alignment has also been formatted.

Reviewer 2 Report

Line 170 - Escherichia coli, in italics and lower case.

I do not understand what data this study wants to provide, what is new or confirmatory. I do not see that it is related to the theme of the magazine to which it has been referred.

..

Author Response

Comments and Suggestions for Authors (Reviewer 2)

Line 170 - Escherichia coli, in italics and lower case.

A: Thank you for pointing this out, this has been correctly formatted at line 208.

I do not understand what data this study wants to provide, what is new or confirmatory. I do not see that it is related to the theme of the magazine to which it has been referred.

A: Thank you for taking the time to review our manuscript. We appreciate your feedback and comments. We have carefully considered your suggestions and comments regarding the scope and aim of the journal and the special issue. However, we respectfully believe that our manuscript is well-aligned with both the scope and aim of the journal, which focuses on qualitative and quantitative research exploring the determinants of antimicrobial use and resistance, as well as the special issue that aims to deepen our understanding of factors related to the misuse of antimicrobials and strategies to improve this misuse in public health. We have also reached out to the guest editors to gauge their interest in our work for the special issue and were pleased to provide their support. We hope that our manuscript will continue to meet your expectations as we move through the review process. Our research explores the determinants of antimicrobial use and resistance, which we believe adds significant value to the field. We have also included an extensive discussion on strategies and interventions that have a positive effect in improving the misuse of antimicrobials, which we hope will contribute to the growing body of knowledge in this area.

Reviewer 3 Report

Thank you for conducting this research. It is interesting indeed.

I have few comments:

1- Being an international study, the number of responses seems very low and can be considered a limitation. I do not agree that internet access could be the reason as you mentioned in the manuscript.

2- As an answer to question 12 in the survey, I think UTI includes (in many cases) E. coli infection. Please re-check the classification of infections you made to make it more accurate.

3- You asked about the spectrum of the usually prescribed antibiotics without asking about the name or family of the antimicrobials particularly. If you have the data, kindly report them according to WHO AWaRe classification. If you don't, please mention this as a limitation.

Author Response

Comments and Suggestions for Authors (Reviewer 3)

Thank you for conducting this research. It is interesting indeed.

A: Thank you for this comment. We would like to overall thank the reviewer for taking the time to go over our manuscript and provide useful and constructive feedback.

I have few comments:

1- Being an international study, the number of responses seems very low and can be considered a limitation. I do not agree that internet access could be the reason as you mentioned in the manuscript.

A: Thank you for this comment. We agree that internet access is not directly relevant to the limitation of the study, however while it is true that internet access is becoming more widespread, it is nonetheless a bias that cannot be ignored. In our study, only people with internet access were able to participate in the survey, and this excludes a significant proportion of the population who do not have access to the internet. We acknowledge that this is a limitation of our study, and we have discussed this in our manuscript.

2- As an answer to question 12 in the survey, I think UTI includes (in many cases) E. coli infection. Please re-check the classification of infections you made to make it more accurate.

A: “Escherichia Coli infections” was a free answer that was given from a survey respondent when asked what the most common infection were managed in pregnant women. This has been clarified in the manuscript, lines 208-211.

3- You asked about the spectrum of the usually prescribed antibiotics without asking about the name or family of the antimicrobials particularly. If you have the data, kindly report them according to WHO AWaRe classification. If you don't, please mention this as a limitation.

A: Thank you for your comment and bringing this to our attention. Unfortunately, we do not have specific information on the names or families of the antimicrobials used by the participants in our study. However, we appreciate your suggestion to report the data according to the WHO AWaRe classification. We included this in the limitation section of the manuscript.

Round 2

Reviewer 2 Report

.